# Diving into Self-Evolving Training for Multimodal Reasoning

**Wei Liu** [* 1] **Junlong Li** [* 1 2] **Xiwen Zhang** [3] **Fan Zhou** [2] **Yu Cheng** [4] **Junxian He** [1]

## Abstract

Self-evolving training—where models iteratively learn from their own outputs—has emerged as a key approach for complex reasoning tasks, addressing the scarcity of high-quality chain-of-thought data. However, its effectiveness in multimodal reasoning, a domain more intricate than text-only reasoning, remains underexplored, and the understanding of critical factors in this training paradigm remains limited. Furthermore, a central challenge for this training method is performance saturation, which impedes further improvements and scalability. Inspired by reinforcement learning (RL), in this paper, we reframe self-evolving training for multimodal reasoning through the lens of RL, identifying three pivotal factors: *Training Method*, *Reward Model*, and *Prompt Variation*. Through systematic analysis, we establish relatively optimal design principles that significantly enhance multimodal reasoning capabilities. Moreover, delving deeper into training dynamics, we uncover the roots of saturation and propose a new automatic balancing mechanism to mitigate this limitation. Building on these insights, we propose M-STAR (**M**ultimodal **S**elf-evolving **T**raining for **R**easoning), a framework that achieves consistent performance gains across models of varying sizes and diverse benchmarks. All resources are made publicly available at https://mstar-lmm.github.io.

## 1. Introduction

Multimodal reasoning is a fundamental skill in many real-world applications, such as intelligent agents (Liu et al., 2024c), robotics (Li et al., 2023; Liu et al., 2024b), and

autonomous driving (Yang et al., 2023). It requires Large Multimodal Models (LMMs) to understand various modalities beyond text. For example, visual mathematical reasoning (Lu et al., 2023) challenges models to analyze complex figures, diagrams, and charts, leveraging the provided information to perform reasoning tasks.

Despite the critical role of mutlimodal reasoning, the availability of human-annotated thought processes in multimodal scenarios remains limited, hindering the learning of multimodal reasoning. Consequently, self-evolving training, which utilizes model's own generation ability to iteratively tune and improve itself without external annotated data, has emerged as an appealing candidate to facilitate reasoning abilities. While research on self-evolving training has primarily focused on the text-only settings (Hosseini et al., 2024; Sun et al., 2024; Shao et al., 2024), its application in the multimodal domain, especially for reasoning tasks, has been limited with only a few sporadic examples (Fang et al., 2024; Dubey et al., 2024; Deng et al., 2024), and a unified framework has yet to be established.

Inspired by reinforcement learning (RL), in this paper, we reframe self-evolving training through the lens of RL, identifying three factors that are critical inside self-evolving training: the **training method**, the use of **reward model**, and **prompt variation**. Through massive controlled studies, we (1) propose a continuous self-evolving training scheme to reduce the gap towards full online learning and outperforms other iterative baselines (§3.2); (2) train the first multimodal, process-based reward model for multimodal reasoning and demonstrate its usefulness in further enhancing performance (§3.3); and (3) find that adding more unlabeled queries helps only when having perfect reward signals (e.g., the oracle groundtruth answers), and it hurts the performance if the reward model does not generalize well on unseen data (§3.4). Beyond static design principles, we investigate the training dynamics, revealing how performance saturation stems from diminishing exploration potential during training. To address this, we introduce a new metric that bridges exploration and exploitation, and propose an automatic balancing mechanism that dynamically adjusts the sampling temperature to sustain exploration-exploitation trade-offs.

Combining all the recipes concluded through separate, controlled studies, we propose our self-evolving training al-

---
[*]Equal contribution [1]The Hong Kong University of Science and Technology [2]Shanghai Jiao Tong University [3]Helixon Research [4]The Chinese University of Hong Kong. Correspondence to: Wei Liu, Junxian He <wliucn, junxianh@cse.ust.hk>, Junlong Li <lockonlvange@gmail.com>.

*Proceedings of the $42^{nd}$ International Conference on Machine Learning*, Vancouver, Canada. PMLR 267, 2025. Copyright 2025 by the author(s).

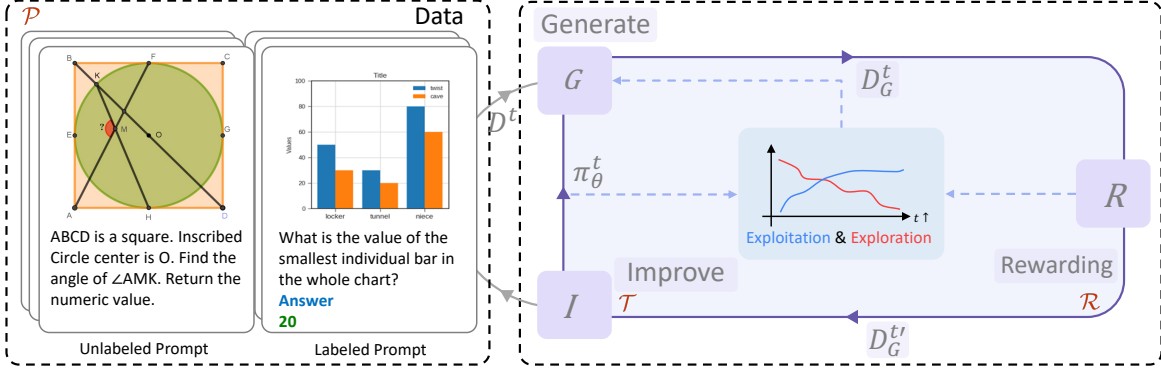

Figure 1: Overview of our self-evolving training framework for multimodal reasoning. We investigate the three essential design components of it, namely Training method ($\mathcal{T}$), Reward model ($\mathcal{R}$), and Prompt variation ($\mathcal{P}$). Orthogonal to the static factors, the Dynamics of self-evoloution is also monitered, and provides control signals to the training process.

gorithm named as M-STAR (**M**ultimodal **S**elf-evolving **Tra**ining for **R**easoning). Our experimental results on 5 different multimodal reasoning benchmarks, including Math-Vista, M3CoT, MMStar, MMBench and AI2D, show that this strategy, which incorporates both optimized static design choices and dynamic adjustments, effectively mitigates exploration loss during training and enhances performance universally for models with varied sizes such as MiniCPM-V-2.5 (8B), Phi-3.5-Vision (4B) and InternVL2 (2B).

## 2. Overview of Self-Evolving Training for Multimodal Reasoning

Self-evolving training can be modeled as a general framework of reinforcement learning, where various algorithms can be formulated as a specific instantiation of RL, such as PPO (Schulman et al., 2017), STaR (Zelikman et al., 2022), ReST (Gulcehre et al., 2023) and ReST$^{EM}$ (Singh et al., 2023). Specifically, given a reward function $\mathcal{R}$, the objective of self-evolving training is to train the policy model $\pi_\theta$ to maximize expectation of reward $\mathcal{R}$:

$$\pi_\theta = \mathrm{argmax}_{\pi_\theta} \sum_i^L \mathbb{E}_{x,o\sim\mathcal{D},\hat{y}_i\sim\pi_\theta[\cdot|x,o]}[\mathcal{R}(\hat{y}_i)], \quad (1)$$

where $x$, $o$ represent the query and image in the given training data $\mathcal{D}$, while $\hat{y}_i$ is a response sampled from the current policy model $\pi_\theta$. This standard RL objective, however, can be unstable to optimize and difficult to scale up, thus a popular algorithm adopted by recent works is to decouple the response rollout $\hat{y}_i \sim \pi_\theta[\cdot|x,o]$ and policy improvement into separate offline stages (Gulcehre et al., 2023; Singh et al., 2023): (1) *Generate*: the current policy model generates new responses $\hat{y}_i \sim \pi_\theta[\cdot|x,o]$; and (2) *Improve*: using the rewards to selects certain responses from the Generate step, which are then used to train the policy model with a standard supervised fine-tuning (SFT) loss. This way, the algorithm

resembles Rejection Fine-Tuning (RFT, Yuan et al. (2023)) as it filters out negative responses in a hard manner. Both steps are performed iteratively to strike a tradeoff between offline and online training. In many tasks such as mathematical problem-solving, there exists a unique, ground-truth answer $a^*$ which is utilized in the reward function, for example, Singh et al. (2023) directly adopts exact match to compute a binary reward by comparing $\hat{y}$ and $a^*$. In such an iterative training procedure, the objective at iteration $t$ is to obtain an improved policy model $\pi_\theta^{t+1}$:

$$\pi_\theta^{t+1} = \mathrm{argmax}_{\pi_\theta^t} \sum_i^L \mathbb{E}_{x,o,a^*\sim\mathcal{D},\hat{y}_i\sim\pi_\theta^t[\cdot|x,o]}[\mathcal{R}(a^*,\hat{y}_i)],$$
$$(2)$$

where the ground-truth answer $a^*$ can be empty, for example, when dealing with unlabeled inputs, the reward model must be able to score $\hat{y}_i$ independently.

**The Design Spaces** Through the lens of Eq. 2, we can identify three dominant factors that influence the training process, (1) *training method*: the training algorithms to perform this iterative process vary as well. For example, while Gulcehre et al. (2023); Xu et al. (2024b) initialize the model from the last checkpoint at each iteration, Zelikman et al. (2022); Singh et al. (2023) argue that initializing from the beginning checkpoint reduces overfitting and gives better performance empirically. (2) *reward model*: the design of reward function $\mathcal{R}$. (3) *prompt variation*: whether to incorporate additional unlabeled inputs without $a^*$ into training. Next, we investigate these three design spaces, aiming to summarize the best practices for each factor.

## 3. Diving into Self-Evolving Design Components

In this section, we explore the three key components of self-evolving training, examining various strategies within each.

We begin by outlining the general setup (§3.1), followed by a comprehensive analysis of each component to identify the best practices for multimodal self-evolution (§3.2-§3.4).

## 3.1. General Setup

**Models** We base our main exploration on MiniCPM-V-2.5 (8B) (Yao et al., 2024), and we also validate the final design choice for each component on two extra models with different sizes: Phi-3.5-Vision (4B) (Abdin et al., 2024) and InternVL-2 (2B) (Chen et al., 2024c). The details of these models can be found in Appendix A. To make the analysis process easier to understand, we mainly present the results of MiniCPM-V-2.5 in this section, while we include the results of the other models in §4.2.

**Datasets** We utilize MathV360K (Shi et al., 2024), a high-quality and diverse multimodal reasoning dataset as our seed training dataset. Specifically, we downsample half of the examples (180K) from it to serve as our labeled training set, while setting aside the remaining half as a unlabeled training set by not using the answers in it. For evaluation, we split 750 samples from the unlabeled part of MathV360K as the in-domain (ID) testset. For our out-of-domain (OOD) testset we use the `testmini` split of MathVista (Lu et al., 2023), a comprehensive benchmark encompassing a wide range of multimodal reasoning tasks, including visual question answering, figure-based question answering, science question answering, and more. We also keep an non-overlapping 250 samples from MathV360K as the global validation set in training.

**Main Training Settings** Before self-evolving training, a self-warmup stage (Appendix B) is applied to enhance the CoT generation ability of all models as many LLMs tend to output final answers directly without CoT. We adopt most of the training settings from Yao et al. (2024) (see Appendix C), using a constant learning rate of $1e-6$ and training for 10K steps across all experiments. During all rollout phases in training, we sample 16 responses per query and set the sampling temperature to 1.0. Unless explicitly stated otherwise, we follow existing practices (Singh et al., 2023; Zelikman et al., 2022) and only use the labeled training data.

## 3.2. Training Methods

As described in §2, there are multiple variants on how we would train to update the policy model. Previous works mainly vary the model initialization factor, where at the "Improve" step, the model can be initialized from either the last checkpoint (Xu et al., 2024b; Pang et al., 2024) or the beginning checkpoint before the first iteration (Zelikman et al., 2022; Singh et al., 2023). Besides model initialization, in this work, we introduce new variants of iterative self-evolving through delving into the gap between iterative training and online RL – concretely, when the iteration

interval is small, the checkpoint at each iteration is initialized from one from the last iteration, and the optimizer as well as the learning rate scheduler is inherited between iterations, then iterative training becomes an online RL algorithm. Therefore, we propose **Continuous Self-Evolving**, a new iterative self-evolving training variant that represents a smoother interpolation between iterative training and online training. In continuous self-evolving training, we inherit the optimizers as well as the learning rate schedulers from the last iteration besides inheriting the model checkpoint, so that the optimization is continuous and closer to purely online learning algorithms. This way, we only have a global optimizer and learning rate scheduler essentially across the entire iterative training process. We also analyze the *iteration interval* effect in continuous self-evolving, which is defined as the training queries passed for one iteration – we specifically study the effect of having a shorter iteration interval, which stands in contrast to the common practice that adopts a long iteration interval to process all the data queries for one iteration.

**Setup** We perform controlled experiments to study the effect of different training methods, thus in this experiment we use the labeled dataset only and simply adopt the binary exact-match reward between ground-truth answer $a^*$ and the generated answer. We compare with the most common iterative self-evolving algorithms ReST$^{EM}$ (Singh et al., 2023) and iterative RFT, which are specific instantiations of our training methods design space. To study the effect of iteration interval in the proposed continuous self-evolving, we experiment with different percentage of all the queries per iteration, varying from [6.25%, 12.5%, 25%, 50%, 100%].

**Results** Table 1 presents the experimental results of various training methods. Overall, initializing training from the last policy model checkpoint $\pi_\theta^t$ and maintaining a continuous optimization process contribute most significantly to the effectiveness of self-evolving training, particularly on MathVista. Continuous self-evolving achieves the best performance both on the in-domain MathV360K test set, with 43.1%, and on the OOD test set, MathVista, with 57.2%. We also see the importance of maintaining a proper interval to traverse the data queries. With a large interval, the training method becomes closer to an offline one, and the model cannot get timely updates on data matching its current output distribution. On the other hand, switching over the *Improve* and *Generate* steps too frequently makes the learning process unstable, leading to a lower score, especially on the in-domain test set. The strategy of continuous self-evolving with proper intervals also works for other smaller models, as shown in Table 6 compared with representative baselines, indicating its effectiveness and generalizability across different model sizes.

Table 1: Accuracy results (%) of self-evolving training using various training methods and iteration intervals. Interval (#) stands for iteration interval, the ratio of data we traverse in one iteration, and we also record the number of corresponding queries. $\mathcal{M}$ represents the policy model from which training is initialized in each iteration. $\mathcal{O}$ denotes whether the optimization process is continuous, i.e., the optimizer states and lr scheduler are inherited from the last checkpoint. Please refer to Table 6 to check the full results on all subtasks of MathVista.

| Method | $\mathcal{M}$ | $\mathcal{O}$ | Interval (%) | MathV360K | MathVista |
|---|---|---|---|---|---|
| MiniCPM-V-2.5 | - | - | - | 13.6 | 52.4 |
| +warmup | - | - | - | 38.8 | 52.6 |
| SFT | - | - | - | 44.3 | 54.8 |
| Iterative RFT | $\pi_\theta^t$ | × | 100% | 42.3 | 55.7 |
| Rest$^{EM}$ | $\pi_\theta^0$ | × | 100% | 42.3 | 55.1 |
| Continous Self-Evolving | $\pi_\theta^t$ | ✓ | 100% | 42.2 | 56.7 |
| | | | 50% | **43.1** | 56.2 |
| | | | 25% | **43.1** | **57.2** |
| | | | 12.5% | 42.3 | 56.1 |
| | | | 6.25% | 41.0 | 56.8 |

## 3.3. Reward Models

In self-evolving training, the most common approach to reward function design uses a binary reward $\mathcal{R}(\hat{y}_i) = \mathbb{1}(\hat{a}_i = a^*)$, where $\hat{a}_i$ is the predicted answer inside $\hat{y}_i$ and incorrect responses are filtered out to maximize rewards. While effective, this sparse binary reward has limitations. It overlooks the quality of the intermediate reasoning steps within a response. Additionally, reward models trained from equal or higher capacity models than the policy model (Fried et al., 2022; Wang et al., 2024; Sun et al., 2024) can provide richer signals to improve the policy model's learning.

In this section, we introduce a Process Reward Model (PRM) (Lightman et al., 2023; Wang et al., 2024) for multimodal reasoning—the first of its kind, to our knowledge—and explore how integrating PRM can enhance reward design and whether it can improve policy model learning in self-evolving training for multimodal reasoning. To incorporate the reward scores into the objective of self-evolving training, the reward function is reformulated as:

$$\mathcal{R}(\hat{y}_i) = \mathcal{H}(\mathbb{1}(a^* = \hat{a}_i) \times \mathcal{R}_p(\hat{y}_i)) \qquad (3)$$

$$\mathcal{R}_p(\hat{y}_i) = \min(f(s_i^0), f(s_i^1), ..., f(s_i^m)) \qquad (4)$$

Here, $\mathcal{H}$ is an operation that processes responses based on the final reward scores, where we ensure all responses are correct by matching the ground truths, and $\mathcal{R}_p(\hat{y}_i)$ represents the process reward score for each sampled response. The function $f(s_i^k)$ denotes the reward score at each intermediate step. Following Lightman et al. (2023), we use the min operation to aggregate stepwise rewards.

**Setup** We conduct controlled experiments to assess the impact of incorporating the Process Reward Model (PRM) into self-evolving training and explore how best to utilize it. Notably, before applying PRM, responses are pre-filtered based on their final answers to ensure consistency and quality during training. To train our PRM, we use Monte Carlo rollouts starting from prefixes with partial reasoning steps (Wang et al., 2024) to generate the training data. Specifically, we sample 16 responses per question and complete each step 8 times to obtain step-level annotations and more details can be found in Appendix D. We evaluate two different $\mathcal{H}$ operations: (1) Top-K: Pick the top-K correct responses according to their reward scores, and (2) Filtering by a Threshold $\alpha$: Filtering out sampled responses with lower aggregated rewards than $\alpha$. The optimal value of $\alpha$ is set 0.2 by grid searching on the validation set. Additionally, we investigate how varying the value of K in Top-K affects training, as it represents a trade-off between the quality and diversity of the samples. According to §3.2, we fix training methods as continuous self-evolving with 45k interval and set continuous self-evolving, with or without randomly selected correct responses as our baselines.

**Results** Table 2 presents the results of integrating the PRM into self-evolving training, along with the impact of different $\mathcal{H}$ choices. Continuous Self-Evolving with PRM using Top-2 achieves the best performance in both the ID and OOD tests, with scores of 45.3% and 59.2%, respectively. Compared to training without PRM, most instances of self-evolving training with PRM show improved performance, especially in the OOD test. Interestingly, randomly selecting a subset of correct responses actually leads to worse performance than continuous self-evolving, suggesting that even correct answers can be noisy. Random selection may increase the proportion of these noisy samples, undermining the effectiveness of self-evolving training.

In terms of leveraging PRM, we found that using Top-K to select the a fixed number of best K responses with high-quality intermediate steps outperforms threshold-based filtering. The results also highlight the importance of balancing the quality and diversity of sampled responses. Selecting K = 2 strikes this balance well, ensuring both response diversity and high-quality reasoning steps for each question. Similar to the results in §3.2, we also see improvement when involving PRM on smaller models in Table 6, Appendix H.

**What makes PRM work for self-evolving training?** To pursue deeper insights into the role of PRM in self-evolving training, we conduct an analysis presented in Figure 2. Based on the results from §3.3, we explore PRM's impact from two key perspectives: (1) Can PRM help the model to select out correct responses among multiple rollouts? (2) How different are the Top 2 and the rest correct solutions re-ranked by reward scores? We use the first checkpoint

Table 2: The results of self-evolving training with PRM and different strategies to leverage reward scores. $\mathcal{H}$ is the method to further pick out high-quality responses from the correct rollouts: (1) Top-k is to select K correct responses with highest rewards, and (2) $> \alpha$ is to pick out the correct responses with rewards larger than $\alpha$. Please refer to Table 6 to check the full results on all sub-tasks of MathVista.

| Method | $\mathcal{H}$ | PRM | MathV360K | MathVista |
|---|---|---|---|---|
| Cont. Self-Evolving | - | × | 43.1 | 57.2 |
| + Random | Random-2 | × | 41.0 | 55.5 |
| +PRM-based Selection | $> \alpha$ | | 43.8 | 57.5 |
| | Top-1 | | 43.0 | 59.0 |
| | Top-2 | ✓ | **45.3** | **59.2** |
| | Top-4 | | 44.0 | 58.4 |

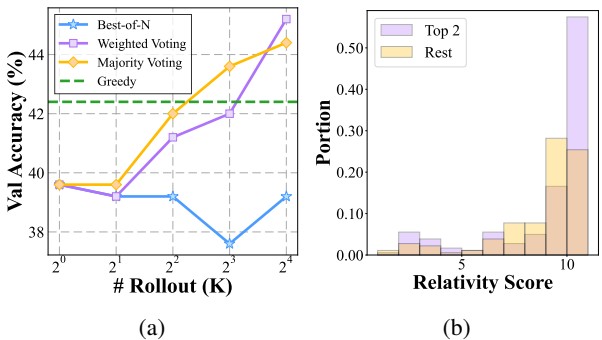

(a)          (b)

Figure 2: **(a)**: Accuracy on the val. set of greedy decoding and three selection strategy across different numbers of rollouts; **(b)**: Average relativity score annotated by GPT4-o of Top 2 and the rest responses re-ranked by rewards, we only calculate on correct ones.

after warmup $\pi_\theta^0$ as policy model to sample 16 responses for each question in the validation set with temperature=1.0 and reveal the behaviors of PRM in these samples.

We evaluate the verification ability of our PRM using two metrics, Best-of-N (BoN) and weighted voting (Sun et al., 2024), which are commonly employed to assess the performance of reward models. Surprisingly, as shown in Figure 2a, our PRM underperforms in both metrics. Notably, BoN and weighted voting yield worse results than vanilla majority voting when $N < 16$. We speculate that this is due to the lack of high-quality step-level annotations compared to text-only reasoning tasks. These findings suggest that our PRM is not an effective **verifier**.

To understand why our PRM can still significantly contributes to self-evolving training despite its weaker verification abilities, we analyzed the distribution of other metrics for the top-2 selected responses compared to other correct responses. We approached this from two perspectives: the average number of reasoning steps, and how much a response is directly relevant to the question (see Appendix E), since we do not find incorrect steps but find some irrelevant steps after randomly checking some examples. We found the responses re-ranked by our PRM generally have **fewer reasoning steps** (Figure 6 in Appendix F) and **more relevant to the query** (Figure 2b). This highlights the **precision** of our PRM in recognizing genuinely high-quality responses. Therefore, our PRM acts as an effective **reranker** to identify top-quality responses. This is especially critical when responses are already filtered by ground-truth answers, and the ability to accurately assess the quality of reasoning steps beyond accuracy becomes vital.

In addition to the aforementioned analysis, we also investigate why leveraging $\alpha$ to filter responses with lower reward scores performs worse than Top-K. The results indicate that, even with the optimal threshold value determined from the validation set, it tends to either retain or filter out all responses for each query, which reduces diversity and makes the learning process more challenging. This further supports the conclusion that **our PRM performs better as a Reranker than as a Verifier**.

### 3.4. Prompt Variation

In this section, we explore how prompt variation affects self-evolving training. There are two primary types of prompts: labeled prompts and unlabeled prompts. Labeled prompts come with annotated ground truth answers, which can be used to filter out incorrect responses during training. In contrast, utilizing unlabeled prompts in self-evolving training is more challenging due to the absence of ground truth annotations. To maintain the quality of unlabeled prompts in training, surrogates like reward scores or pseudo labels must be employed. Meanwhile, unlike labeled prompts, unlabeled prompts are not be trained in SFT period, which increases the difficulty of learning for policy models.

**Skylines: Unlabeled Prompts with Oracle Reward Signals** The coupling of these additional factors introduces complexity, making the effective use of unlabeled prompts less predictable. Therefore we start by establishing a baseline with "skyline" experiments, where both the unlabeled prompts and their ground truth answers are available but not used during the SFT phase. These unlabeled prompts with oracle reward signals serve as an intermediate difficulty between fully unlabeled and labeled prompts, providing insight into the challenges of training with unlabeled data.

**Unlabeled Prompts** We incorporate unlabeled prompts into self-evolving training. To ensure the quality of sampled responses for these prompts, we use weighted voting to ensemble the predictions from different responses, treating the ensembled prediction as a pseudo label $\tilde{a}$. This pseudo label is then used to filter out responses with conflicting

Table 3: Results of involving unlabeled data. $T_{\texttt{mixin}}$ denotes when to mixin the unlabeled data. The use of PRM follows §3.3, except we first get a pesudo "ground truth" through weighted voting on unlabeled prompts.

| Oracle | PRM | $T_{\texttt{mixin}}$ | MathV360K | MathVista |
|---|---|---|---|---|
| - | × | - | 43.1 | 57.2 |
| - | ✓ | - | 45.3 | 59.2 |
| ✓ | × | 0% | 42.5 | 58.2 |
| ✓ | ✓ | 0% | 42.9 | 59.1 |
| × | ✓ | 0% | 43.3 | 58.2 |
| × | ✓ | 25% | 42.4 | 57.6 |
| × | ✓ | 50% | 42.9 | 58.2 |
| × | ✓ | 75% | 45.0 | 58.8 |

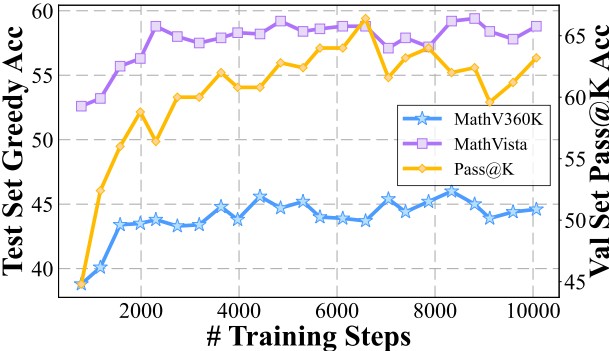

Figure 3: Opposite trend of Greedy Accuracy and Pass@K.

self-evolving training can harm the process, potentially causing a deviation in the policy model's distribution.

predictions, ensuring consistency. Following the best practices outlined in §3.3, we apply PRM as a reranker to select the top-2 responses with the predicted answer $\tilde{a}$. These unlabeled prompts are then mixed with labeled ones for self-evolving training. Additionally, since learning from unlabeled prompts is more challenging for policy models, we investigate the optimal timing to introduce them into training as well. We maintain an interval of 45k prompts and adjust when unlabeled prompts are introduced into the training process. Specifically, we introduce unlabeled prompts after [0%, 25%, 50%, 75%] of the total training process.

**A Glimpse at Unlabeled Prompts: Potential Efforts to Make Them Effective** Table 3 presents the results of incorporating unlabeled prompts with and without oracle reward signals. When training relies solely on oracle reward signals without the PRM, continuous self-evolving with unlabeled prompts outperforms standard continuous self-evolving trained only on labeled prompts in the out-of-domain test but underperforms in the in-domain test. This indicates that additional prompts help the model generalize better to underrepresented questions but also increase the risk of forgetting previously learned information. However, after combining with our PRM, all policy models perform worse than our best model trained exclusively on labeled prompts in both benchmarks, even when oracle reward signals are provided. Based on the analysis in §3.3, this occurs since our PRM is unable to verify responses without ground-truth answers, and its generalization remains a concern.

When examining the timing for introducing unlabeled prompts, we find that adding them from the beginning helps mitigate the negative impact on model performance, compared to introducing them in midway. However, when unlabeled prompts are introduced later in the training process, they participate less in the overall training, leading to better results simply due to their limited involvement. This suggests that, without sufficient surrogate supervision (e.g., precise reward signals), introducing unlabeled prompts into

## 4. Dynamics of Self-Evolution & Final Recipe

So far, we have explored the impact of three pivotal factors within our design space, leading to established best practices for learning multimodal reasoning – we adopt continuous self-evolving training coupled with a reward model to help data selection as described in §3.3, and we perform the training process on SFT datasets with final answer annotations. In this section, we delve even deeper into the current self-evolution strategy to better understand the bottlenecks. Instead of analyzing from a design space perspective as previously, we now fix the design parameters and focus exclusively on the training dynamics during the model's self-evolution. This shift in focus allows us to examine the process from an orthogonal angle, providing further insights into the underlying mechanisms that drive or impede progress in multimodal reasoning capabilities.

### 4.1. Monitoring the Training Dynamics

Intuitively, two critical conditions must be met for the success of self-evolving training: (1) the presence of high-quality candidate responses generated by the model, otherwise self-evolving will not work no matter how strong the reward is; and (2) the reward function's ability to effectively distinguish and prioritize these high-quality responses. These conditions align with the traditional reinforcement learning concepts of *exploration* and *exploitation*. Apparently, both exploration and exploitation capabilities are dynamic targets in self-evolving training, as the policy model evolves and the distribution of rollout responses changes with each iteration. To better understand these training dynamics, we propose tracking and visualizing three metrics, where we introduce a novel metric, Reward-Pass@2, to monitor the exploration-exploitation trade-offs:

- *Greedy Accuracy*: the model's accuracy with greedy decoding. We track this metric for reference to compare

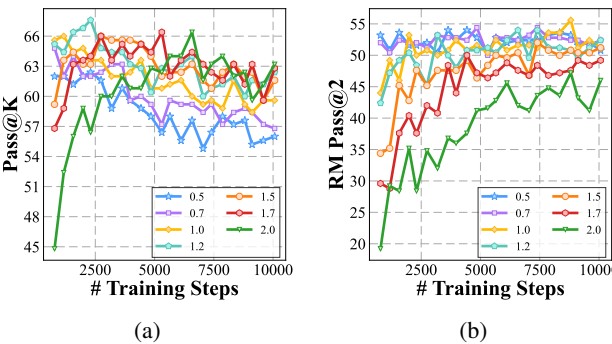

(a)                                    (b)

Figure 4: **(a)**: Pass@K decreases for all different temperatures; **(b)**: The Reward-Pass@2 saturates quickly. All metrics are calculated on validation set.

with other metrics.

- *Pass@K Accuracy*: the percentage of samples for which the model produces at least one correct response when sampling $K$ candidates. This metric measures the model's exploration ability.

- *Reward-Pass@2*: the ratio (%) of samples for which there exist correct responses among the top 2 responses ranked by the reward model. This metric directly reflects the exploitation efficacy of the reward model for the current policy. We choose Pass@2 since our training strategy involves selecting the top 2 responses using the reward model (§3.3).

Specifically, after each training iteration of our current optimal strategy, we sample 16 responses from the model checkpoint on the validation set, with the temperature range set to $t = [0.5, 0.7, 1.0, 1.2, 1.5, 1.7, 2.0]$. We analyze with varying temperatures as temperature is a key hyperparameter for the generation diversity and model's exploration.

**Results** Figure 3 shows a clear trend where, as training progresses, the Pass@K metric continuously declines while greedy accuracy improves. This pattern indicates the loss of exploration ability, which hampers the model's potential for continuous improvement and may lead to performance saturation. These observations are consistent with findings in text-only settings as reported by Wu et al. (2024). In Figure 4a we analyze Pass@K accuracy at various temperatures and observe a significant trend: despite a general decay in exploration ability, larger temperatures tend to resist this decline more effectively, allowing the model to maintain a stronger ability to explore in the mid to late stages of training. This observation suggests that the optimal temperature for training may need to be dynamically adjusted throughout the self-evolving process, rather than being fixed at the outset as is currently common practice.

In Figure 4b, we observe that the Reward-Pass@2 metric initially increases but quickly reaches a plateau, indicating that

Table 4: Results on MathVista with various training strategies across multiple model sizes. We highlight the relative improvement of M-STAR over the pre-evolved model, i.e., the "+warmup" row. CPM = MiniCPM-V-2.5, Phi = Phi-3.5-vision, and Intern = InternVL2-2B.

| Variant | CPM | Phi | Intern |
|---|---|---|---|
| **Base** | 52.4 | 46.5 | 46.4 |
| **+warmup** | 52.8 | 49.3 | 47.6 |
| **SFT** | 54.7 | 49.5 | 41.9 |
| **Iterative RFT** | 55.7 | 50.2 | 47.5 |
| Rest$^{EM}$ | 55.1 | 50.5 | 47.9 |
| **Cont. Self-Evolving** | 57.2 | 51.1 | 48.4 |
| **+ PRM Re-Rank** | 59.2$_{\uparrow 6.4}$ | 53.2$_{\uparrow 3.9}$ | 48.8$_{\uparrow 1.2}$ |
| **M-STAR (*Reward-Pass@2*)** | 59.5$_{\uparrow 6.7}$ | 54.5$_{\uparrow 5.2}$ | 50.3$_{\uparrow 2.7}$ |

the reward model's capacity to exploit further diminishes as training progresses. This limitation could be due to both the reduced exploration ability and the inherent constraints of the reward model. Next, we fix the reward model as a control variable and ask, **how can we enhance exploration to allow the reward model to exploit more effectively?**[1]

### 4.2. M-STAR– Final Recipe with Optimal Design Choices & Adaptive Explorations

Reward-Pass@2 closely relates to the effectiveness of our self-evolving training strategy since our method selects top responses ranked by the reward model, and Reward-Pass@K directly reflects the quality of these 2 responses.[2] While Reward-Pass@2 naturally measures exploitation when the policy is fixed, the absolute value of this metric actually encapsulates both exploration and exploitation – its value would be low if the model fails to explore high-quality candidates. Therefore, we hypothesize that enhancing the Reward-Pass@K scores for the current iteration through varied configurations could potentially improve the efficacy of self-evolving training. We fix reward model as a control variable and focus on modifying the model's exploration capabilities to achieve this objective. Analysis in §4.1 suggests that the temperature, which is crucial for exploration, may require dynamic adjustment. Thus we propose to adjust the temperature automatically at each iteration based on the validation Reward-Pass@2 scores. This aims to optimize exploration so that the selected responses are of higher quality, potentially enhancing overall training effectiveness.

---

[1]While improvements to the reward model could also enhance Reward-Pass@2, we reserve it for future work.

[2]We note that there is a slight mismatch between Reward-Pass@2 and our training strategy, as we pre-filter responses using the ground-truth answer before the reward model reranks them. Ideally, a more aligned metric would measure the CoT reasoning quality of the top 2 responses, both containing correct answers. Given that there is no reliable method to score the quality of the thought processes, we consider Reward-Pass@2 as a reasonable approximation which turns out to be effective empirically.

Table 5: Performance of M-STAR compared with baselines. We highlight the relative improvement of M-STAR over the pre-evolved model, i.e., the "+warmup" row. For benchmark with suffix "-R", we follow Xu et al. (2024a) to remove some perception sub-tasks in them, to get the subsets that focus more on reasoning.

| | MathVista | M3CoT | MMStar-R | MMBench-R | AI2D | Average |
|---|---|---|---|---|---|---|
| MiniCPM-V-2.5 | 52.4 | 41.2 | 44.6 | 72.6 | 64.4 | 55.0 |
| + warmup | 52.6 | 47.8 | 45.1 | 76.9 | 65.9 | 57.7 |
| M-STAR | **59.5**$_{\uparrow 6.9}$ | **48.7**$_{\uparrow 0.9}$ | **50.7**$_{\uparrow 5.6}$ | **79.9**$_{\uparrow 3}$ | **69.1**$_{\uparrow 3.2}$ | **61.6**$_{\uparrow 3.9}$ |
| Phi-3.5-vision | 46.5 | 39.4 | 42.5 | 56.8 | 47.5 | 46.5 |
| + warmup | 49.3 | 46.5 | 44.2 | 70.9 | 65.5 | 55.3 |
| M-STAR | **54.5**$_{\uparrow 5.2}$ | **51.3**$_{\uparrow 4.8}$ | **48.8**$_{\uparrow 4.6}$ | **73.6**$_{\uparrow 2.7}$ | **67.9**$_{\uparrow 2.4}$ | **59.2**$_{\uparrow 3.9}$ |
| InternVL2-2B | 46.4 | 16.7 | 20.0 | 14.2 | 33.5 | 26.2 |
| + warmup | 47.6 | 45.6 | 41.8 | **68.8** | **60.0** | 52.8 |
| M-STAR | **50.3**$_{\uparrow 2.7}$ | **47.1**$_{\uparrow 1.5}$ | **42.0**$_{\uparrow 0.2}$ | 67.3$_{\downarrow 1.5}$ | 59.7$_{\downarrow 0.3}$ | **53.3**$_{\uparrow 0.5}$ |

Specifically, we adjust the temperature per two iterations, and pick the temperature from 0.3 to 1.6 with interal 0.1 automatically with maximum validation Reward-Pass@2 scores. The optimal design choices outlined in §3, combined with our adaptive exploration strategy, form our final recipe for multimodal self-evolving training for reasoning, M-STAR. For experiments on Phi-3.5-vision and InternVL2, considering the limited capacity of these models and the computational cost, we utilized both the warmup data and multimodal PRM based on MiniCPM-V-2.5.

**Full Results** Table 4 presents the results of our final approach as well as the comparison with representative baselines. We also demonstrate the scores on all sub-tasks of MathVista in Table 6, Appendix H. We see that by incorporating the dynamics of Reward-Pass@2, which balances both exploration and exploitation, our final recipe achieves the highest results for all three backbone LMMs. In addition to overall trend, we observe that self-evolving training based on larger models yields more comprehensive improvements. We assume that the smaller model like InternVL2-2B may struggle to generalize its learned abilities across different domains as effectively as the larger models, such as MiniCPMV-2.5 and Phi-3.5-vision.

We also plot how the Pass@K and Reward-Pass@2 change for M-STAR when trained on MiniCPM-V-2.5. To align with training, we show the metrics corresponding to the selected temperature in each iteration (see Appendix G for others). Figure 5 shows that compared with choosing a fixed temperature over the whole process statically, tuning it automatically mitigate the regression of Pass@K to avoid the exploration loss. Besides, the Reward-Pass@2 is also generally higher than before. These further highlight the necessity to monitor the dynamics and adjust accordingly.

**M-STAR on More Diverse Benchmarks** To further investigate how well M-STAR generalizes to multiple benchmarks, we select four extra multi-modal ones focus on reasoning as well: M3CoT (Chen et al., 2024b), MMStar

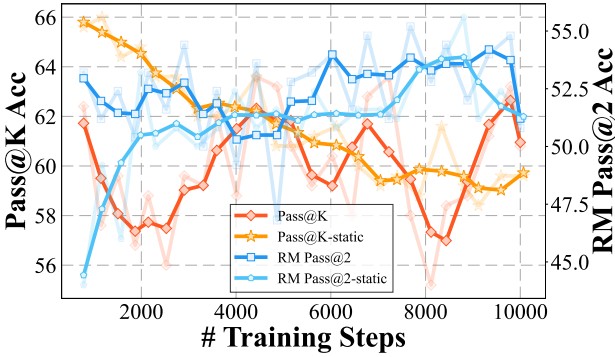

Figure 5: Comparing the smoothed Pass@K and Reward-Pass@2 curves with the optimal static training progress, which fixs temperature $T = 1.0$.

(Chen et al., 2024a), MMBench (Dev set, v1.1) (Liu et al., 2025), AI2D (Kembhavi et al., 2016). For MMStar and MMBench, we remove the perception sub-tasks to construct subsets focus more on reasoning. As shown in Table 5, models self-evolved with M-STAR consistently outperform both the base models and those trained with warmup across nearly all benchmarks. The only exception is InternVL2-2B, which underperforms on two benchmarks, aligning with the speculations discussed above. Smaller models face challenges in generalizing beyond their training data, particularly on perception-intensive benchmarks like MMBench-R and AI2D. In contrast, larger models such as Phi-3.5-vision and MiniCPM-V-2.5 show significantly improved generalization, despite being trained with the same query set.

## 5. Conclusion

We dive into the self-evolving training for multimodal reasoning. Three static components are identified at first, namely the training method, reward model and the prompt variation. Through controlled experiments, we conclude a set of optimal design choices. On the other direction, we also go deeper into the dynamics of self-evolving training to

analyze the trade-off between exploitation and exploration. By balancing the training dynamics, we are able to further improve its performance. We hope our work can provide insights and guidance for future research on self-evolving training for multimodal reasoning.

## Acknowledgments

This project is partially supported by NSFC Grant 62306177.

## Impact Statement

This paper presents work whose goal is to advance the field of Machine Learning. There are many potential societal consequences of our work, none which we feel must be specifically highlighted here.

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

## A. Details of Selected LMMs

**MiniCPM-V-2.5 (Yao et al., 2024)**   is a powerful, openly released LMM. MiniCPM-V-2.5 leverages LLaMA-3-8B (Meta, 2024) for its language model and SigLIP (Zhai et al., 2023) as its vision encoder, resulting in strong multimodal capabilities. Its performance on a wide range of multimodal benchmarks significantly surpasses previous openly released LMMs such as LLaVA (Liu et al., 2023; 2024a) and Qwen-VL (Bai et al., 2023).

**Phi-3.5-Vision (Abdin et al., 2024)**   is a multimodal model combining a CLIP ViT-L/14 (Radford et al., 2021) image encoder and a Phi-3.5-mini transformer decoder. It processes interleaved image-text inputs using dynamic cropping for images and is pre-trained on 0.5T tokens from diverse datasets. Post-training via supervised fine-tuning (SFT) and direct preference optimization (DPO) enhances its multimodal reasoning and language understanding capabilities.

**InternVL-2 (Chen et al., 2024c)**   InternVL 2.0 is a multimodal large language model series ranging from 1B to 108B parameters. The specific 2B version we use combines InternViT (300M) (Chen et al., 2024c), an MLP projector, and InternLM-2-Chat (1.8B) (Cai et al., 2024), showcasing strong vision-language capabilities. Built with progressive alignment training, it efficiently aligns vision and language models while supporting diverse inputs (text, images, video, medical data) and outputs (images, bounding boxes, masks), performing competitively across various vision-language tasks.

## B. Warm-Up Phase to Unlock the Chain-of-Thought (CoT) Capability of LMMs

In our preliminary experiments, we found that open-source LMMs would directly output the answer given the query, while struggling to produce detailed chain-of-thought (CoT) reasoning processes. This may originate from the the scarcity of high quality rationales in most existing multimodal SFT training datasets (Masry et al., 2022; Shi et al., 2024), which limits the ability of open-source LMMs to generate detailed, step-by-step reasoning. Self-evolving training, however, requires responses with varying intermediate steps to allow models to learn effectively from on-policy data. To address this issue, we initiate a warm-up phase to collect some CoT data from the model itself as the first step before self-evolving training. Instead of prompting the model to answer questions directly, we prompt it to generate intermediate reasoning steps for a given triplet (question, image, and answer) using the following instruction:

> **Extra instruction to guide CoT**
>
> Offer a comprehensive breakdown of your analytical process, detailing each step, the reasoning behind your decisions, and how you integrated various pieces of information, and put your answer at the end.

For each triplet, we ask models to rollout 16 samples with temperature $= 1.0$. We then filter out results where the final answers do not match the ground truth and sample 100K from the generated dataset to create a warm-up CoT dataset $\mathcal{D}_w$ with correct answers. Finally, we fine-tune our models on this dataset, treating it as a standard RFT process. Our iterative self-evolving training process will then start from this model checkpoint after the warm-up training.

## C. Hyper Parameters

We follow the training setup from Yao et al. (2024), using a learning rate of 1e-6 and a batch size of 128. A constant learning rate scheduler with a warmup ratio of 0.1 is applied. Input images are encoded using SigLIP SoViT-400m/14 (Zhai et al., 2023), and the visual tokens are compressed through a perceiver resampler structure with a single cross-attention layer. Additionally, each input image is sliced into a maximum of 9 segments, with each segment compressed into 96 queries.

## D. Training Process Reward Model (PRM)

To train our PRM, we first train another checkpoint (denoted as $\hat{\pi}_\theta^0$) on our CoT-augmented training data for a much longer period to make sure it fully converges.

Based on this model, we leverage Monte Carlo Rollut method (Wang et al., 2024) to collect the training data for PRM. Specially, we randomly pick 50K questions from the full training set, and sample 16 responses for each of them with $\hat{\pi}_\theta^0$. We de-duplicate these responses, and only keep at most 4 responses for each question. After that we randomly sample 50K question-response pairs from all the pairs, where we control the ratio of correct and wrong responses as 1:1, and the ratio of

multi-choice and free-form question as 1:1 as well, to keep a balanced distribution.

To construct the labels of each step, we use $\hat{\pi}_\theta^0$ as the completer to complete the solution from the end of each step in one response. For the $k^{\text{th}}$ step, the step label is annotated as $\frac{1}{N}\sum_{j=1}^{N}\mathbb{1}(C_j(s^{\leq k})=a^*)$, where $N(=16)$ is the number of completion, $C_j$ is the $j$-th completion.

Based on the stepwise annotations, we train our PRM from $\hat{\pi}_\theta^0$. We initialize the linear reward model head as the average of the embeddings, and train with MSE loss on all tokens, where the label of each token is identical to the step end token. In experiments we freeze the visual encoder as we find it brings a slight improvement.

## E. Measuring Response Relativity

To get a comprehensive understanding of how our PRM works as a re-ranker, we conduct a quantitative analysis using GPT4-o (`gpt-4o-2024-08-06`) to see how much a correct response is directly related to the query, e.g., does not contain irrelvant steps. The prompt we use is as follows:

> **Prompt for GPT4-o to annotate the relativity score**
>
> Given the image and a related question, you need to judge how a candidate solution is directly related to the question. You need to consider all its steps, and return a final value bewteen 1-10 as a overall score.
> Conclude your judgement at the end as "So the relativity score is X" where X is the score you give.
>
> [Question]
> {question}
>
> [Solution]
> {solution}

## F. Extra Comparison between RM-selected Solutions and Others

As a complement to our analysis in §3.3, we additionally plot the number of reasoning steps (split by \n\n) for the top-2 solutions re-ranked by our process reward model, as well as for the other solutions. We observe that the top-2 solutions typically have fewer reasoning steps, indicating that the reward model can effectively identify solutions with fewer irrelevant steps and prioritize more straightforward ones.

## G. More Results for M-STAR

We plot the extra analysis results for M-STAR here. In Figure 7, we plot the changes of Pass@K and Reward-Pass@2 across different temperatures for M-STAR(*Reward-Pass@2*) as a compliment to the adapative adjustion mentioned in §4.2. We can see that acroos all selected temperatures, the exploration ability reflected by Pass@K does not regress continuously, and the Reward-Pass@2 reaches its peak more quickly, compared with training without the monitor of dynamics.

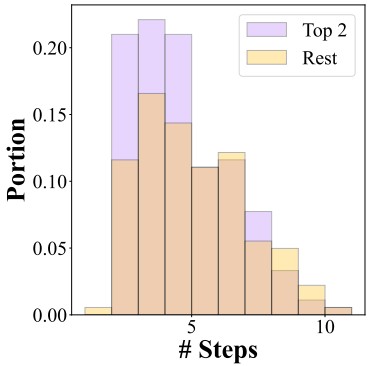

Figure 6: Number of reasoning steps for top-2 solutions selected by our reward model and the rest solutions.

## H. Full Results of MathVista

We present the complete evaluation results on MathVista for all three of our selected models. The scores for each subtask in MathVista are reported in Table 6. As shown, M-STAR achieves improvements across all subtasks compared to the pre-evolved models, particularly on geometric problems and math word problems. This demonstrates its enhanced comprehensive multimodal reasoning ability across multiple aspects.

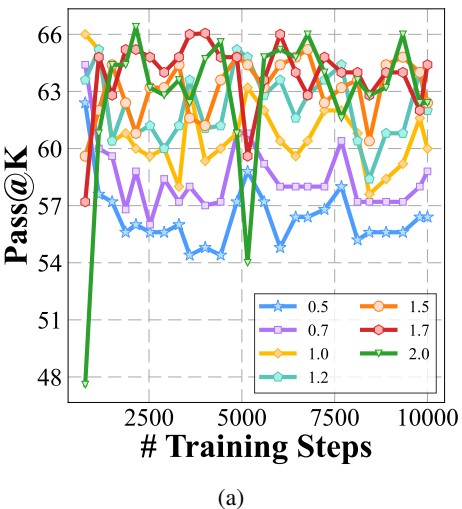
(a)

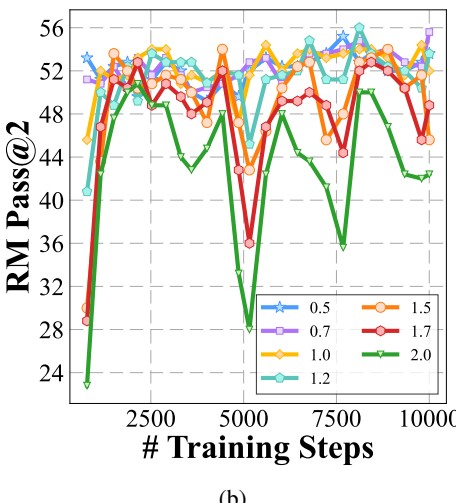
(b)

Figure 7: **(a)**:Pass@K changes during the training of M-STAR (*Reward-Pass@2*); **(b)**: :Reward-Pass@2 changes during the training of M-STAR (*Reward-Pass@2*). We pick 7 different temperatures.

Table 6: Full analysis of MathVista. Task types: FQA: figure question answering, GPS: geometry problem solving, MWP: math word problem, TQA: textbook question answering, VQA: visual question answering. We highlight the relative improvement of M-STAR over the pre-evolved model, i.e., the "+warmup" row.

| Model | ALL | FQA | GPS | MWP | TQA | VQA |
|---|---|---|---|---|---|---|
| *MiniCPMV-2.5* | | | | | | |
| MiniCPMV-2.5 | 52.4 | 59.2 | 44.7 | 50.5 | 53.8 | 48.0 |
| +warmup | 52.8 | 58.4 | 47.1 | 57.0 | 53.8 | 45.8 |
| SFT | 54.7 | 58.7 | 50.5 | 56.5 | 55.7 | 50.8 |
| Iterative RFT | 55.7 | 59.1 | 49.5 | 65.6 | 55.1 | 48.0 |
| Rest$^{EM}$ | 55.1 | 58.0 | 49.5 | 64.5 | 55.1 | 47.5 |
| Cont. Self-Evolving | 57.2 | 57.6 | 56.3 | 65.1 | 57.0 | 49.7 |
| +PRM Re-Rank | $59.2_{\uparrow 6.4}$ | $59.1_{\uparrow 0.7}$ | $\mathbf{61.1}_{\uparrow 14}$ | $\mathbf{68.3}_{\uparrow 11.3}$ | $55.1_{\uparrow 1.3}$ | $51.4_{\uparrow 5.6}$ |
| M-STAR (*Reward-Pass@2*) | $\mathbf{59.5}_{\uparrow 6.7}$ | $\mathbf{59.5}_{\uparrow 1.1}$ | $59.1_{\uparrow 12}$ | $65.6_{\uparrow 8.6}$ | $\mathbf{58.9}_{\uparrow 5.1}$ | $\mathbf{54.2}_{\uparrow 8.4}$ |
| *Phi-3.5-vision* | | | | | | |
| Phi-3.5-vision | 46.5 | $\mathbf{58.7}$ | 36.5 | 36.0 | 56.3 | 41.9 |
| +warmup | 49.3 | 55.8 | 42.8 | 53.2 | 55.1 | 38.0 |
| SFT | 49.5 | 53.9 | 52.9 | 52.7 | 49.4 | 35.8 |
| Iterative RFT | 50.2 | 58.4 | 41.4 | 50.0 | 55.7 | 43.0 |
| Rest$^{EM}$ | 50.5 | 56.8 | 46.6 | 49.5 | $\mathbf{58.9}$ | 39.7 |
| Cont. Self-Evolving | 51.1 | 56.1 | 48.6 | 55.9 | 52.5 | 40.2 |
| +PRM Re-Rank | $53.2_{\uparrow 3.9}$ | $56.9_{\uparrow 1.1}$ | $51.9_{\uparrow 9.1}$ | $60.8_{\uparrow 7.6}$ | $55.1_{0}$ | $39.7_{\uparrow 1.7}$ |
| M-STAR (*Reward-Pass@2*) | $\mathbf{54.5}_{\uparrow 5.2}$ | $56.9_{\uparrow 1.1}$ | $\mathbf{56.7}_{\uparrow 13.9}$ | $57.5_{\uparrow 4.3}$ | $55.1_{0}$ | $\mathbf{44.7}_{\uparrow 6.7}$ |
| *InternVL2-2B* | | | | | | |
| InternVL2-2B | 46.4 | 53.2 | 45.2 | 33.3 | 50.0 | $\mathbf{48.0}$ |
| +warmup | 47.6 | 52.4 | 54.8 | 46.2 | 43.7 | 36.9 |
| SFT | 41.9 | 37.5 | 40.4 | 49.5 | 32.3 | 50.8 |
| Iterative RFT | 47.5 | 49.8 | $\mathbf{57.7}$ | 52.1 | 41.8 | 32.4 |
| Rest$^{EM}$ | 47.9 | 49.4 | 54.8 | 51.1 | $\mathbf{51.3}$ | 31.3 |
| Cont. Self-Evolving | 48.4 | $\mathbf{53.2}$ | 50.5 | 56.5 | 40.5 | 37.4 |
| +PRM Re-Rank | $48.8_{\uparrow 1.2}$ | $52.0_{\downarrow 0.4}$ | $55.8_{\uparrow 1}$ | $52.1_{\uparrow 5.9}$ | $45.6_{\uparrow 1.9}$ | $35.2_{\downarrow 1.7}$ |
| M-STAR (*Reward-Pass@2*) | $\mathbf{50.3}_{\uparrow 2.7}$ | $49.4_{\downarrow 3}$ | $57.2_{\uparrow 2.4}$ | $\mathbf{65.0}_{\uparrow 18.8}$ | $42.4_{\downarrow 1.3}$ | $35.2_{\downarrow 1.7}$ |

# I. More Related Works

In this section we will briefly introduce other works that are related to our study that cannot be elaborated in the main context due to page limit.

**Self-Evolving Methods**  The most straightforward and widely-used approach to enhance a model's reasoning ability is through supervised fine-tuning (SFT), where models mimic the outputs of highly capable models (Yu et al., 2023; Yue et al., 2023). However, as the gap between open-source models and proprietary ones narrows, the performance improvements from SFT tend to plateau. This has led to increased attention on self-evolving methods, where models refine and improve themselves without external supervision, as a means to further boost their reasoning abilities.

Some early self-evolving approaches primarily focus on single-round improvements. For instance, LMSI (Huang et al., 2022) leverages CoT prompting combined with self-consistency to generate high-confidence solutions from unlabeled data, which are then used to augment the training process. Similarly, RFT (Yuan et al., 2023) enriches the training data by filtering solutions using existing labels. Both methods apply this augmentation process in just one iteration.

On the other hand, several works have explored iterative approaches for self-improvement. Notably, STaR (Zelikman et al., 2022), ReST$^{EM}$ (Singh et al., 2023), and V-STaR (Hosseini et al., 2024) retrain their models from the original checkpoint after each iteration, while RAFT (Dong et al., 2023), ReST (Gulcehre et al., 2023) and ReST-MCTS$^*$ (Zhang et al., 2024a)continuously fine-tune models starting from the previous iteration's checkpoint. Reinforcement Learning (RL) techniques also fit into this iterative category, offering an online mechanism that tightly couples exploration and exploitation. RL methods, such as PPO (Schulman et al., 2017) and GRPO (Shao et al., 2024), are frequently applied to unlabeled data, using an additional reward model to evaluate the quality of generated responses. GRPO, in particular, streamlines the process by removing the value model from PPO and instead leverages in-batch comparison to estimate advantages across different rollouts, providing a more stable alternative.

**Multimodal Reasoning**  Currently, the most common approach to improve multimodal reasoning capabilities continues to be supervised fine-tuning (SFT). For example, G-LLaVA (Gao et al., 2023) augments existing geometry question-answering datasets to fine-tune the LLaVA-1.5 model (Liu et al., 2024a). Math-LLaVA (Shi et al., 2024) selects and augments data from larger multimodal question-answer datasets, carefully balancing the difficulty of samples. Similarly, MAVIS (Zhang et al., 2024b) focuses on the geometry and function domains and generates instruction-based tuning data through synthetic data engines.

However, recent works have begun incorporating self-evolving mechanisms into multimodal reasoning. For instance, VILA$^2$ (Fang et al., 2024) iteratively improves its image captioning performance by generating increasingly detailed captions, which are subsequently used to retrain the model. LLaMA3-V (Dubey et al., 2024) employs a reject-sampling strategy to generate missing explanations for question-answer pairs that lack intermediate reasoning steps in existing multimodal datasets, thereby enhancing the model's reasoning capabilities.

**Connecting to Recent Rule-based RL**  Recent work in online RL with rule-based rewards has demonstrated strong performance on complex reasoning tasks like MATH (DeepSeek-AI et al., 2025; Zeng et al., 2025a), suggesting that using PRMs in GRPO training may constrain improvement. While these approaches leverage rule-based rewards to enhance the reasoning capabilities of (M)LLMs, our M-STAR framework is compatible with these findings for two key reasons. First, our reward strategy is also rule-based: we filter out responses with incorrect answers before applying PRM, effectively using PRM as a reranker to select trajectories with the highest-quality reasoning steps. Second, our training setup differs significantly from approaches like DeepSeek-R1. While R1 applies policy gradients at every step using PRM (as in GRPO), which can introduce reward hacking issues, our STaR-like (Sun et al., 2024; Zeng et al., 2025b) method avoids step-wise policy gradients. Although RL with rule-based rewards provides excellent performance, we believe exploring how to effectively train PRMs tailored to more general scenarios (Chae et al., 2025), and integrating PRMs into online RL training, remain valuable directions for achieving even higher performance ceilings.

