# OpenReview forum: "Diving into Self-Evolving Training for Multimodal Reasoning"
_ICML.cc/2025/Conference — ICML 2025 poster_

### Official Review · Reviewer_HhcC · 2025-03-09

**Overall Recommendation:** 3

**Summary:**

This paper investigates self-evolving training for multimodal reasoning through the lens of reinforcement learning, identifying three key factors: Training Method, Reward Model, and Prompt Variation. The authors propose a continuous self-evolving training scheme that inherits optimizer states between iterations, develop a multimodal Process Reward Model (PRM) for reranking responses, and analyze the impact of unlabeled data. They also examine training dynamics, identifying performance saturation as a key challenge, and propose an automatic balancing mechanism to adjust sampling temperature. These components are combined into a framework called M-STAR (Multimodal Self-evolving Training for Reasoning) which is evaluated across multiple multimodal reasoning benchmarks using different model sizes. The authors report consistent performance improvements, particularly on MathVista where their approach achieves a 6.7% absolute improvement over the pre-evolved model.

## update after rebuttal
The authors address most of my questions, e.g. PRM as effective reranker, and statistical significances of results. I still have concerns about the effectiveness of the self-improving methods as the improvements are small in most benchmarks. I am still on the fence, slightly more positive. So I increase my rating. AC, please note that, I think the paper is borderline.

**Claims And Evidence:**

The paper makes several claims that are not fully supported by convincing evidence:

1. The authors claim their PRM is effective as a reranker despite not good as a verifier. While they show some analysis in Figure 2, the explanation lacks depth and rigor. The paper notes that PRM-selected responses have fewer reasoning steps and are more relevant to queries, but doesn't provide a compelling theoretical or empirical explanation for why this makes PRM effective in the self-evolving training context.

2. The improvements reported on benchmarks are modest (mostly 1-6% absolute gains), and it's not clear how significant these improvements are statistically.

3. The paper claims that continuous self-evolving training with proper intervals is better than traditional approaches, but the performance differences shown in Table 1 are small (less than 3% in many cases, e.g. 57.2% vs 55.1%), raising questions about the significance of this contribution.

**Essential References Not Discussed:**

ReFT: Reasoning with Reinforced Fine-Tuning
Trung Quoc Luong, Xinbo Zhang, Zhanming Jie, Peng Sun, Xiaoran Jin, Hang Li

**Experimental Designs Or Analyses:**

The experimental design has several issues:

1. The analysis of why PRM works as a reranker despite being ineffective as a verifier is insufficient. This is a critical insight that could be valuable to the field, but the paper doesn't explore it deeply enough.

2. The experiments on training dynamics are interesting but preliminary.

**Methods And Evaluation Criteria:**

The methods and evaluation criteria are generally appropriate for the problem at hand. The authors use established multimodal reasoning benchmarks and conduct controlled experiments to isolate the impact of different components. However, there are some issues:

1. The authors use GPT4-o to measure "response relativity," but don't provide sufficient details about the reliability of this metric or potential biases in this evaluation approach.

2. The paper makes a case of PRM. This seems to be opposite to the findings in the DeepSeek-R1 paper (though the paper is on text based reasoning tasks). Therefore, this deserves more in-depth evaluation.

**Other Comments Or Suggestions:**

- The paper would benefit from a clearer articulation of its novel contributions relative to prior work.

- A more rigorous analysis of why PRM works as a reranker despite being ineffective as a verifier would significantly strengthen the paper.

- The paper should address recent findings that question the effectiveness of PRMs for reasoning tasks and explain why their approach might overcome these limitations.

**Other Strengths And Weaknesses:**

Strengths:
- The paper provides a systematic exploration of different components of self-evolving training for multimodal reasoning.
- The introduction of continuous self-evolving training with inherited optimizer states is a potentially useful contribution.
- The analysis of training dynamics and the proposed automatic balancing mechanism addresses an important challenge in self-evolving training.

Weaknesses:
- The paper lacks novelty in its core components. Most of the techniques have been explored in prior work, and the paper doesn't clearly articulate what is fundamentally new.
- The improvements over baselines are modest, raising questions about the practical significance of the approach.
- The most intriguing finding—that PRM can be effective as a reranker despite being ineffective as a verifier—is not explored deeply enough to provide meaningful insights.
- The paper doesn't adequately address the limitations of PRMs for reasoning tasks that have been identified in recent literature.

**Questions For Authors:**

1. Your paper shows that your PRM is ineffective as a verifier but effective as a reranker. This is an intriguing finding that contradicts conventional wisdom about PRMs. Can you provide a more rigorous analysis or theoretical explanation for why this is the case? How does this relate to recent findings in papers like DeepSeek-R1 that question the effectiveness of PRMs for reasoning tasks?

2. The improvements reported on benchmarks are relatively modest (mostly 1-6%). Have you conducted statistical significance tests to ensure these improvements are meaningful? How do these improvements compare to the variance observed across different training runs?

3. How does your approach specifically differ from prior work like STaR, ReST, and ReST^EM beyond the application to multimodal reasoning? What are the novel technical contributions that distinguish your work?

I am open to upgrade my rating if important questions are addressed or clarified satisfactorily.

**Relation To Broader Scientific Literature:**

The paper positions itself within the self-evolving training literature but doesn't clearly differentiate its contributions from prior work. The authors mention related approaches like STaR, ReST, and ReST^EM, but don't provide a detailed comparison of how their approach differs technically or conceptually.

The paper also doesn't adequately address recent findings that question the effectiveness of PRMs for reasoning tasks, such as those mentioned in the DeepSeek-R1 paper, which found that PRMs have limitations in guiding reasoning tasks due to challenges in defining fine-grained steps and assessing their correctness.

**Theoretical Claims:**

The paper makes limited theoretical claims, primarily framing self-evolving training as a reinforcement learning problem. The formulation appears sound, but the paper doesn't develop this theoretical foundation into novel insights that significantly advance our understanding of self-evolving training.

---

> ### Author Rebuttal · Authors · 2025-04-01
>
> Thanks for your reivew.
>
> # Q1
>
> > 1. ...shows that your PRM is…
>
> > 2. How... relate to recent findings in papers like DS-R1…
>
> ## Q1.1 PRM Analysis
>
> First, we want to clarify an important distinction between BoN selection and reranking using PRM.  But due to the space limit, please refer to **Q1 in Response to Reviewer gU79** for details about this **clarification and analyses** below.
>
> To better investigate and validate the behaviour of the PRM, we conducted several analyses, which help better understand the PRM and our findings.
>
> **Value Function Analysis**:
>
> PRM serves as a value estimator trained via Monte Carlo rollouts. We show our PRM is with a lower MSE for correct responses (0.081) than incorrect ones (0.124), supporting its strength in reranking valid reasoning paths over verifying incorrect ones.
>
> **Human Evaluation**:
>
> We validate GPT-4o’s relativity score (Figure 2) via human annotation across 50 examples, showing ~84% agreement and confirming its reliability as an alignment measure.
>
> **Readability Analysis**:
>
> Using the Flesch Reading Ease score, we find that top-2 reranked responses are more readable and less erratic, indicating PRM’s benefit in producing clearer, more coherent outputs.
>
> **Qualitative Case Study**:
>
>
> Even when final answers are correct, PRM prefers responses with more coherent reasoning, illustrating its advantage in identifying high-quality rationales beyond binary correctness.
>
>
> ## Q1.2
> Thanks for the insightful question!
> 1. As discussed in **Q1.1**, our use of PRM plays a distinct role in our framework. First, our reward strategy also follows rule-based—we filter out responses with incorrect answers before using PRM. In this way, PRM is used more like a reranker, helping us select trajectories with the highest-quality reasoning steps that are also consistent with the original question.
> 2. Second, the training setup differs significantly from R1. While in DeepSeek-R1’s discussion, it applies policy gradients at every step using PRM (as in GRPO), which can introduce reward hacking issues, our method is STaR-like and avoids this risk by not using step-wise policy gradients.
> 3. While PRMs have higher ceilings, rule-based rewards are more robust and stable—one reason for their success in large-scale setups like R1. Still, we believe exploring PRMs further is valuable for pushing performance even higher.
>
> # Q2
>
> > The improvements reported…. Have you conducted statistical significance…? How do these improvements…?
>
> ## Q2.1
>
> In the context of self-evolving or RL-based methods on reasoning tasks, 3–6% absolute gains are substantial. Prior work such as RestEM reports similar improvements in the 1–6% range. Importantly, our benchmarks span diverse subtasks, and we observe much larger improvements on some of individual benchmarks, for instance, a notable 12% absolute gain on geometry problem-solving. These improvements are consistent across different model sizes and benchmarks.
>
> To validate the statistical significance of our results, we perform t-test on the predictions, to assess whether the hypothesis that “MiniCPMV with M-STaR in Table 4 is better than Cont. Self-Evolving + PRM-Rerank on MathVista” is statistically significant or not. Our results demonstrate that the result is significant with a **p-value < 0.042**.
>
> Regarding stability, we observe a variance of around **0.04** across three independent runs of our static optimal strategy, which is acceptable given the computational constraints. Hence, we do not perform extensive repeated runs in later experiments due to expensive computation requirements.
>
> ## Q2.2
>
> > The paper claims … in Table 1 are small (less than 3% in many cases, e.g. 57.2% vs 55.1%),...
>
> As mentioned in **Q2.1**, continuous self-evolving training is one part of our overall framework and not expected to drive all performance gains alone. Similar to prior work (e.g. Tulu-2.5), changes to the training algorithm typically yield modest yet meaningful improvements in reasoning tasks.
>
> Nonetheless, optimising each component is crucial to fully realising the potential of the complete M-STaR framework. We conduct a significance t-test specifically for cont self-evolving training and Iterative RFT in Table 1 on MathVista, which results in a **p-value < 0.02**—proving the significance.
> And the improvement across different models is also consistent according to Table 4.
>
> # Q3
>
> > How does your approach differ …? …novel technical…?
>
> We will clarify the distinctions between our approach and prior STaR-like methods in the related work section. Briefly, each component of our framework introduces a new design aimed at improving and advancing beyond traditional STaR-like approaches—such as continuous self-evolving training, using PRM as a reranker, dynamic self-evolution, and their integration into a unified framework for multimodal reasoning.
> These components did not exist in the mentioned previous works, and they lead to significant gains as we reported in the submission.

---

### Official Review · Reviewer_dYgr · 2025-03-14

**Overall Recommendation:** 3

**Summary:**

This paper identifies three key components in multimodal reasoning models that require further exploration. It systematically analyzes and unveils the critical aspects of training methods, reward models, and prompt design. Additionally, it proposes the use of appropriate temperature adjustment to balance exploration and exploitation. The final approach is validated through extensive experiments.


## update after rebuttal
Thank the authors for the clarifications. I would like to keep my current rating.

**Claims And Evidence:**

Most claims made in the submission are supported by experimental evidence.

Line 175-178  “...switching over the Improve and Generate steps too frequently makes the learning process unstable, leading to a lower score, especially on the in-domain test set.” lacks of evidence.

**Essential References Not Discussed:**

The paper is self-contained.

**Experimental Designs Or Analyses:**

I did check all the experimental designs and analyses.

**Methods And Evaluation Criteria:**

I believe that the methods and corresponding evaluation metrics in this paper are well-aligned and detailed.

**Other Comments Or Suggestions:**

1 Please re-organize the first paragraph in Section 3.2 to make sure that “iteration interval” is defined first before being used to avoid unnecessary confusion.

2 “…we fix training methods as continuous self-evolving with 45k interval” Please specify which iteration interval the 45k interval refers to. [6.25%,12.5%,25%,50%,100%]

**Other Strengths And Weaknesses:**

- The paper is well-written and easy to follow.

- This paper systematically presents the key factors for improving model performance.

**Questions For Authors:**

1 In Table 6, continuous evolving performs worst in MiniCPMV2.5 FQA task, can the authors explain this result?

2 Also in Table 6, for MiniCPMV2.5, continous evolving with PRM + rerank negatively influence a lot on TQA task. Meanwhile, for Phi3.5, continous evolving performs really bad on TQA task but PRM + rerank has a positive influence on this task. What’s the dynamic behind this observation?

3 Can the author explain the motivation to focus on the short iteration interval as mentioned in line 146-149 in section 3.2?

**Relation To Broader Scientific Literature:**

This paper provides practical insights for the multimodal reasoning community, particularly in the design of training methods and process reward models.

**Theoretical Claims:**

N/A

---

> ### Author Rebuttal · Authors · 2025-04-01
>
> Thanks for your review. And we appreciate your recognition of your work. We would address your concerns one by one:
>
> # Q1
>
> > Line 175-178 “...switching over the Improve and Generate steps too frequently makes the learning process unstable, leading to a lower score, especially on the in-domain test set.” lacks of evidence.
>
> Thanks for your question. In our table 1., less interval means switching over the Improve and Generate Steps more frequently, and we can found from the results that too small intervals would hurt the perf in in-domain test set (MathV360K testset)
>
> # Q2
>
> > Please re-organize the first paragraph in Section 3.2 to make sure that “iteration interval” is defined first before being used to avoid unnecessary confusion.
>
> Thanks for your suggestion. We would follow your suggestion to define the “iteration interval” first and further clarify it.
>
> # Q3
>
> > “…we fix training methods as continuous self-evolving with 45k interval” Please specify which iteration interval the 45k interval refers to. [6.25%,12.5%,25%,50%,100%]
>
>
> We are sorry for the confusion. Since the total amount of training data is 180K (line 124), [6.25%,12.5%,25%,50%,100%] correspond to 11K, 22K, 45K, 90K, 180K respectively.
>
> # Q4
>
>  > In Table 6, continuous evolving performs worst in MiniCPMV2.5 FQA task, can the authors explain this result?
>
> > Also in Table 6, for MiniCPMV2.5, continous evolving with PRM + rerank … on TQA task. Meanwhile, for Phi3.5, continous evolving performs really bad on TQA task but PRM + rerank has a positive influence on this task. What’s the dynamic behind this observation?
>
> Thank you for your thoughtful observations regarding the performance dynamics in Table 6. We address both points below in a unified explanation.
>
> - Overall, **M-STaR** consistently outperforms other variants, and significance testing (p < 0.046) supports the robustness of this improvement.
> - Compared to GPS and MWP, which demand more complex multimodal reasoning, tasks like TQA and FQA rely more heavily on perception abilities, such as visual understanding or interpreting structured text layouts. While reasoning is still required, the relative emphasis shifts more toward perception in these tasks.
> - Without dynamic monitoring, the self-evolving training process may saturate, over-optimizing for reasoning ability and neglecting perception skills. This leads to unstable or suboptimal performance on tasks like TQA and FQA.
> - As shown in Table 6, M-STaR effectively mitigates these issues. It adaptively adjusts the evolution process by introducing dynamic control to avoid training saturation. The improvements are especially significant when both the LLM and vision encoder have sufficient capacity, confirming M-STaR’s effectiveness across diverse subtasks.
>
> # Q5
>
> > Can the author explain the motivation to focus on the short iteration interval as mentioned in line 146-149 in section 3.2?
>
> Thank you for your question.
> We focus on short iteration intervals because our study explores optimal training design through the lens of RL. In the context of LLM training, several prior works [1–3] have shown that online training methods with appropriate iteration interval outperform offline ones. In our self-evolving training framework, a shorter iteration interval corresponds to an more online training regime, where the model can adapt more quickly to newly generated samples. Our findings are consistent with existing RL literature, which emphasizes that iteration intervals should be appropriately short—not too large (to avoid stale or offline updates), and not too small (to prevent excessive variance across iterations).
>
> [1] Direct Language Model Alignment from Online AI Feedback ICML2024
>
> [2] DeepSeek-R1: Incentivizing Reasoning Capability in LLMs via Reinforcement Learning
>
> [3] SimpleRL-Zoo: Investigating and Taming Zero Reinforcement Learning for Open Base Models in the Wild

---

> > ### Comment · Reviewer_dYgr · 2025-04-06
> >
> > Thank the authors for clarifications. I would like to keep my current rating.

---

### Official Review · Reviewer_gU79 · 2025-03-14

**Overall Recommendation:** 4

**Summary:**

The paper introduces M-STAR—a framework that reframes self-evolving training for multimodal reasoning as a reinforcement learning (RL) problem. It identifies three critical factors (training method, reward model, and prompt variation) and proposes a continuous self-evolving training variant. A novel Process Reward Model (PRM) is designed to assess intermediate reasoning quality, and an adaptive exploration strategy (via automatic temperature tuning) is introduced to mitigate performance saturation. Extensive experiments across multiple benchmarks (e.g., MathV360K, MathVista) and model sizes (MiniCPM-V-2.5, Phi-3.5-Vision, InternVL2-2B) demonstrate consistent gains.

**Claims And Evidence:**

Claims:

(i) continuous self-evolving training outperforms conventional iterative methods

(ii) integrating a multimodal PRM enhances candidate selection

(iii) adaptive temperature tuning effectively balances exploration and exploitation

Evidence:

The authors support these claims with controlled ablation studies (e.g., Table 1 and Table 2) and analysis of metrics such as Reward-Pass@2 and Pass@K. However, the effectiveness of PRM as a reranker—despite underperformance on standard verification metrics—needs further clarification.

**Essential References Not Discussed:**

No

**Experimental Designs Or Analyses:**

The experimental design is comprehensive, with clear comparisons between various training strategies (iterative, continuous, PRM-enhanced). The analysis of exploration–exploitation dynamics via Reward-Pass@2 is particularly insightful. However, the potential impact of noisy unlabeled data in the PRM setup warrants deeper discussion.

**Methods And Evaluation Criteria:**

The proposed method is well aligned with the multimodal reasoning challenge, specifically addressing the scarcity of high-quality chain-of-thought annotations. The evaluation covers multiple baselines, model sizes, and both in-domain and out-of-domain benchmarks.

**Other Comments Or Suggestions:**

Minor typos (e.g., “mutlimodal” instead of “multimodal” in Line 013 col 2 word 6) should be corrected. Though not critical, some explanations—especially around the adaptive temperature mechanism—could be clarified for enhanced readability.

**Other Strengths And Weaknesses:**

Strengths:

- Novel reformulation of self-evolving training as an RL problem.

- Introduction of continuous training and adaptive exploration, supported by thorough empirical evaluations.

- Detailed analysis of reward model dynamics and exploration–exploitation trade-offs.

Weaknesses:

- Some ambiguity remains regarding the PRM’s role as a reranker given its lower performance on standard verification metrics
- limited diversity in benchmark tasks may restrict claims about generalizability.

**Questions For Authors:**

1. Can you clarify the apparent discrepancy between the PRM’s verification metrics (e.g., BoN, weighted voting) and its effectiveness in reranking responses?

2. How robust is the continuous self-evolving training process to variations in iteration interval and temperature adjustment parameters?

**Relation To Broader Scientific Literature:**

The paper is well-situated within recent work on self-training, multimodal reasoning, and RL-based training methods (e.g., references to Singh et al. (2023)[1], Zelikman et al. (2022)[2], Hosseini, Ali, et al.[3] and related chain-of-thought literature).

[1] Singh, Avi, John D. Co-Reyes, and Rishabh Agarwal. "Beyond Human Data: Scaling Self-Training for Problem-Solving with Language Models." ICLR 2024 Workshop on Navigating and Addressing Data Problems for Foundation Models

[2] Zelikman, Eric, et al. "Star: Bootstrapping reasoning with reasoning." Advances in Neural Information Processing Systems 35 (2022): 15476-15488.

[3] Hosseini, Arian, et al. "V-STaR: Training Verifiers for Self-Taught Reasoners." CoRR (2024).

**Theoretical Claims:**

The reformulation of self-evolving training as an RL objective is sound, and the derivations (e.g., Eq. 1 and Eq. 2) appear correct. While the continuous optimization approach is promising, further details on convergence guarantees and stability—especially during adaptive temperature tuning—would enhance the theoretical contribution.

---

> ### Author Rebuttal · Authors · 2025-03-31
>
> # Q1
>
> > However, the effectiveness of PRM as a reranker—despite underperformance on standard verification metrics—needs further clarification
>
> Thank you for your question.
> First, we would like to clarify an important distinction between Best-of-N (BoN) selection and reranking using PRM. When using PRM for BoN, no ground-truth answers are provided—meaning the PRM must identify both correct and incorrect responses independently. In contrast, during our training (excluding the unlabeled prompt setting), the responses are already **filtered using ground-truth answers**, allowing the PRM to focus on reranking correct responses. We then select the top-2 responses as high-quality samples. This is why the BoN results do not align with our usage of PRM during training.
>
> To better investigate and validate the behaviour of the PRM, we conducted four analyses:
>
> - **Value Function Perspective**:
> Our PRM is trained via MC-rollouts [1, 2], acting as a value function estimating the expected reward (i.e., answer correctness) from an intermediate reasoning step. To assess accuracy, we compare the predicted value score (the PRM score at the step with the lowest value) against an empirical value score (the proportion of 16 rollouts from that step leading to a correct answer).
> We then compute the mean squared error (MSE) between the PRM-predicted value and this empirical value across the dataset, grouped by whether the original response was ultimately correct or incorrect.
>
> |       |  Correct   |  Wrong   |
> |-----|-----|-----|
> |  MSE   |   0.081  |   0.124  |
>
> This result indicates two things.
>
> 1. It highlights why our PRM here is more suitable as a reranker rather than a verifier—while PRM effectively distinguishes good reasoning trajectories for correct responses, its predictions are less reliable for incorrect responses, where it can be more easily confused.
>
> 2. Responses filtered by answers with higher PRM scores are with higher quality, making them better samples to learn.
>
>
> This suggests future evaluations could separate reranking performance (when answers are known) from value estimation (on unlabeled or incorrect responses).
>
> - **Human Evaluation**:
> To validate the reliability of the relativity score presented in Figure 2 in our paper and assess whether GPT-4o is unbiased, we conducted a human evaluation to measure the alignment between questions and responses. Specifically, we invited two human annotators to label 50 responses. To facilitate more consistent judgments, we categorized the relativity score into three levels: not relevant (corresponding to original scores 1–4), somewhat relevant (scores 4–7), and very relevant (scores 8–10). We then computed the agreement between the human annotations and GPT-4o's automatic scores. The average agreement exceeded 84%, which, according to prior research on automatic evaluation [3], indicates a strong level of consistency. This result supports the reliability of both our proposed relativity score and the GPT-4o annotations shown in Figure 2.
>
> - **Readability Analysis**:
> We use the Flesch Reading Ease (FRE) metric, we found that top-2 responses are more consistently readable than other samples with slightly higher mean (64.48 vs. 64.25) and much smaller variance (28.48 vs. 55.97)
>
> This suggests PRM reranking helps reduce incoherent or erratic language, yielding clearer responses overall.
>
> [1] Math-Shepherd: ... ACL2024
>
> [2] ProcessBench
> - **Qualitative Case Study**:
> The stepwise reward provides a finer signal of reasoning quality. Even when answers are correct, reasoning may be flawed. For example:
> ```
> Question: What type of MRI is shown in the image?\nChoices:\n(A) T2-weighted  \n(B) Diffusion-weighted  \n(C) FLAIR  \n(D) T1-weighted.
>
> Top-2 Response: The choice is D, because the image is described as having a \"very bright signal\", ….\n\nDark areas on T1-weighted MRI typically …\n\n# Answer\n\nD
>
> Other Response: The given MRI is indicated as a diffusion-weighted analysis.\n\nTherefore, the correct answer to the problem is: \n\n# Answer\n\nD
> ```
> Although both predict the right answer, the top-2 response has clearer and more consistent reasoning, while the other includes misleading claims—underscoring the value of stepwise evaluation.
>
> # Q2
>
> > limited diversity in benchmark tasks
>
> As shown in Tables 5 and 6, the five benchmarks we used include many diverse subtasks, covering not only the math domain but also tasks such as visual qa, figure qa, logic-qa, scientific reasoning, spatial reasoning, etc, which are very diverse and challenging. And they are also common practices to evaluate multimodal reasoning (e.g. math-llava)
>
> # Q3
>
> > How robust is the continuous self-evolving..
>
> For the robustness of hyperparameters, in table 4, we follow the same parameters as Table 1,2 to train Phi-3 and InternVL models. Considering they are two models with different series and sizes, but the results remain consistent with MiniCPMV, we believe it proves hyperparameters in MSTaR are robust.

---

### Official Review · Reviewer_grRq · 2025-03-15

**Overall Recommendation:** 3

**Summary:**

The authors reframe self-evolving training for multimodal reasoning through the lens of RL and indentify three factors: training method, the use of reward model, and prompt variation. They train the first multimodal, process-based reward model for multimodal reasoning and
demonstrate its usefulness in further enhancing performance; and find that adding more unlabeled queries helps only when having perfect reward signals (e.g., the oracle groundtruth answers), and it hurts the performance if the reward model does not generalize well on unseen data.

**Claims And Evidence:**

The authors reframe self-evolving training for multimodal reasoning through the lens of RL and indentify three factors: training method, the use of reward model, and prompt variation. But the experiments do not deeply study how these three factors effect the final results. (i.e., the ablation studies are not enough, the settings of the ablation studies focus primarily on minor hyperparameter adjustments, leading to conclusions that align with conventional expectations.)

**Essential References Not Discussed:**

NA.

**Experimental Designs Or Analyses:**

1. Why do the authors choose Minicpm, internvl, and phi models as base models? Can the proposed method use in LLaVA and QwenVL? I think these two models are more common in MLLM research.
2. The experiments are not solid enough, the proposed method should conduct on more multimodal reasoning benchmarks.
3. The compared baselines are not enough,  For instance, the author should compare their method with some existing self-evolution methods in LLM and MLLM reasoning methods.
4. Can the proposed method further improve the existing MLLM reasoning methods? Just using a base model (like Minicpm) is not solid.
5. The ablation studies should delve deeper into algorithmic comparisons. For instance, contrasting with techniques like RLHF,  DPO, GRPO, as well as exploring different training methodologies (e.g., multi-training stages), the sequence of training stages.

**Methods And Evaluation Criteria:**

The technical contribution is limited, it seems the authors use some existing LLM  techniques in MLLM. However, what are the new technical challenges induced by MLLM and how do the authors solve them? The paper lacks a hypothesis-driven structure that ties the findings to the central research question, which appears more like a tech report rather than a research paper.

**Other Comments Or Suggestions:**

The related work about Multimodal Reasoning is not enough, the authors should consider more recent papers.

**Other Strengths And Weaknesses:**

1. The paper is well-organized and easy to follow.
2. The research topic: enhancing reasoning in large multimodal models without external annotated data is a good topic.

**Questions For Authors:**

How do you train the PRM? Can you give me a more clear explanation? The data, training method, etc.

What is unlabeled prompt? In line 126, I find the answer is missing. Is it just a math question?
Besides, why do you use the word 'prompt' to describe this answer missing setting, I think is a little bit weird. Why not use label?

In Tab5., what is the training data? Just the math data can improve?

**Relation To Broader Scientific Literature:**

The paper lacks a hypothesis-driven structure that ties the findings to the central research question, which appears more like a tech report rather than a research paper.

**Theoretical Claims:**

Not available.

---

> ### Author Rebuttal · Authors · 2025-03-31
>
> Thanks for your time and effort in reviewing our paper.
>
> # Q1
> > limited technical novelty
>
> Overall we have contributed:
> - A pilot study to enhance multimodal (MM) reasoning validated by comprehensive studies,
> - The first self-evolving training recipe that blended online training and PRMs in MM reasoning
> - The new method, monitoring training dynamics, and the experiments to show the effectiveness of it.
> - Significant improvements on various multimodal MM benchmarks
>
> They span three underexplored areas:
> - A comprehensive RL-based analysis of self-evolving training;
> - A new training framework and PRM tailored to multimodal reasoning;
> - A methodology for tracking and interpreting training dynamics, offering insights into model evolution.
>
> # Q2
> > lack of a hypothesis-driven structure
>
> Our central research question is outlined in the earlier sections of the paper (Abstract, Sec 1 Lines 12-26, 43-48).
>
> While writing structures vary across papers, we adopt a component-wise, empirical exploration to derive insights—a format used in prior works like [1,2]. We believe this structure does not diminish the rigour&clarity of our contributions.
>
> [1] Unpacking DPO and PPO: Disentangling Best Practices for Learning from Preference Feedback, NeurIPS 2024
>
> [2] What Matters When Building Vision-Language Models? NeurIPS 2024
>
> # Q3
> > Why do the authors choose Minicpm, internvl, and phi models ...
>
> We selected them as they were among the strongest open-source LMMs of their sizes (2B, 4B, 7B) at the time of our experiments, offering solid reasoning abilities necessary for self-evolving training.
>
> We prioritized:
> - Models with solid baseline reasoning, critical for RL-based methods
> - Stronger open-source models, as improving them is more meaningful than weaker ones.
>
> Although LLaVA and Qwen-VL are widely used, they underperformed during our study period. For instance, LLaVA-1.6 achieved only ~20–30 on MathVista, and Qwen-VL-Chat lagged behind MiniCPM-V 7B ( ~40 v.s. 52.8).
>
> Our method is model-agnostic and can be extended to these models in future work.
>
> # Q4
> > Evaluation on more MM reasoning benchmarks
>
> In Tables 5 and 6, the five benchmarks we use include many diverse subtasks, covering not only math but also tasks such as visual qa, figure qa, logic-qa, scientific reasoning, spatial reasoning, etc, which are very diverse and challenging. And previous works[1,2] often evaluate on just 1-2 benchmarks
>
> [1] Bootstrapping Mathematical Reasoning for Multimodal Large Language Models EMNLP2024
>
> [2] Mathematical Visual Instruction Tuning with an Automatic Data Engine ICLR2025
>
> # Q5
> > baseline comparisons are insufficient
>
> Our work is the first to systematically apply self-evolving training to MM reasoning. In Table 1, we also include the most widely used self-evolution baselines in text-only settings: Iterative RFT (aligned with STaR [1] and ReST [2]) and RestEM—three of the most established methods at the time of submission.
>
> [1] STaR: Bootstrapping Reasoning With Reasoning Neurips2022
>
> [2] Reinforced Self-Training (ReST) for Language Modeling
>
> # Q6
> > Can the proposed method further improve the existing MLLM reasoning methods? Just using a base model (like Minicpm) is not solid.
>
> As shown in Table 4, we have validated our method on three models of varying sizes, and the conclusions remain consistent. Also, as in many MLLM/LLM reasoning works (e.g. ReST, V-STAR), it's common to mainly conduct experiments on top of general models  rather than stacking different reasoning methods.
>
> # Q7
> > The ablation studies should delve deeper into algorithmic comparisons…
>
> At the time of submission, STaR-like methods were among the most effective and compute-efficient for reasoning tasks, especially in MM contexts, as deployed to develop e.g. Llama3, ReST-MCTS* etc. Other RL paradigms like GRPO were less explored in this setting and are beyond our scope.
>
> # Q8
> > Related Work
>
> We leave related work in **Appendix I**. And we will move them to main body and add more recent papers in the next revision.
>
> # Q9
> > PRM Training Details
>
> We have provided details in **Appendix D**, but briefly: The PRM is trained using via MC-Rollout, where 50K questions are sampled and each is completed with up to 16 responses using a converged model checkpoint. Stepwise annotations are generated based on completion correctness, and the PRM is trained with token-level MSE loss. The dataset is balanced across correct/incorrect responses and question types.
>
> # Q10
> > Unlabeled Prompts
>
> Unlabeled prompts simulate real-world scenarios where collecting answers is difficult. Since PRM is trained on diverse reasoning steps, we test whether it can generalize to unlabeled data—enabling broader scalability beyond labeled datasets.
>
> # Q11
> > Training Data
>
> In line 121 of our paper, we use MathV360K, which includes not only MM math problems, but also a diverse range of tasks like function QA, figure-based QA, and more, which covers a broad spectrum of multimodal reasoning scenarios.

---

### Decision · Program_Chairs · 2025-05-01

**Decision:**

Accept (poster)

**Comment:**

This paper presents an interesting reframing of self-evolving training for multimodal reasoning through a RL lens, identifying key factors such as training method, reward modeling, and prompt variation. The introduction of the first multimodal, process-based reward model is a notable contribution, demonstrating potential for performance enhancement. However, the evaluation lacks depth in exploring the identified factors through sufficient ablation studies and comparisons with existing state-of-the-art methods and benchmarks, particularly on more common LMM architectures. While the findings regarding the impact of unlabeled data based on reward signal quality are insightful, the overall contribution would be strengthened by more rigorous experimentation and a clearer articulation of the novel technical challenges addressed within the multimodal context.